# Genetic Structure and Diversity of Hatchery and Wild Populations of Yellow Catfish *Tachysurus fulvidraco* (Siluriformes: Bagridae) from Korea

**DOI:** 10.3390/ijms25073923

**Published:** 2024-03-31

**Authors:** Kang-Rae Kim, Keun-Yong Kim, Ha Yoon Song

**Affiliations:** 1Department of Life Science & Biotechnology, Soonchunhyang University, Asan 31538, Republic of Korea; kimkangrae9586@gmail.com; 2Department of Genetic Analysis, AquaGenTech Co., Ltd., Busan 48300, Republic of Korea; koby0323@hanmail.net; 3Inland Fisheries Research Institute, National Institute of Fisheries Science, Geumsan 32762, Republic of Korea

**Keywords:** microsatellite, bottleneck, population structure, genetic diversity, *Tachysurus fulvidraco*

## Abstract

Yellow catfish *Tachysurus fulvidraco* is an important commercial fish species in South Korea. However, due to their current declines in its distribution area and population size, it is being released from hatchery populations into wild populations. Hatchery populations also produced from wild broodstocks are used for its captive breeding. We reported 15 new microsatellite DNA markers of *T. fulvidraco* to identify the genetic diversity and structure of its hatchery and wild populations, providing baseline data for useful resource development strategies. The observed heterozygosity of the hatchery populations ranged from 0.816 to 0.873, and that of the wild populations ranged from 0.771 to 0.840. Their inbreeding coefficient ranged from −0.078 to 0.024. All populations experienced a bottleneck (*p* < 0.05), with effective population sizes ranging from 21 to infinity. Their gene structure was divided into two groups with STRUCTURE results of *K* = 2. It was confirmed that each hatchery population originated from a different wild population. This study provides genetic information necessary for the future development and conservation of fishery resources for *T*. *fulvidraco*.

## 1. Introduction

Aquaculture is increasingly becoming a crucial source of protein, with freshwater aquaculture also serving as an important resource [1]. Freshwater ecosystems are critical to maintaining economic significance [2]. However, it is the most vulnerable ecosystem due to pressure from human activities [2]. Yellow catfish *Tachysurus fulvidraco* (Richardson, 1846) is one of the most important farmed freshwater fish species in China and South Korea, with China recording production of 565,477 tons and 587,822 tons in 2021 [3] and 2022 [4], respectively. According to the Fisheries Resources Agency of South Korea, domestic releases of *T. fulvidraco* from 2017 to 2022 amounted to 34,462,711 tons (https://www.fira.or.kr/fira/main.jsp: accessed on 1 May 2022).

In South Korea, fishery for *T. fulvidraco* is very popular today (https://www.fira.or.kr/fira/main.jsp: accessed on 1 May 2022). Commercial fishing of *T. fulvidraco* has continued in wild populations such as the Hantan River, Seomjin River, and Geum River in the Korean Peninsula, but its catches are decreasing due to ecosystem changes. Therefore, to prevent this decline and sustain the catches, the local government has continued to release *T. fulvidraco* by purchasing hatchery populations from private hatcheries after captive breeding of broodstock collected from the wild. In this process, its genetic diversity is an important consideration when collecting broodstocks from wild populations [5]. Release of hatchery-reared individuals can increase fishery productivity, accelerate the recovery of depleted stocks, and ensure stock survival [5]. High genetic diversity affects the hatchability and survival rates of offspring [6]. In this process, if genetic management is not carried out, genetic variation in hatchery fry may be reduced, which may have a negative impact on wild populations after release [7]. Therefore, hatcheries must take genetic diversity into account and collect a variety of broodstock to maintain high genetic diversity [6,7].

Changes in catches of commercially important fish species are primarily caused by diverse human activities, for example habitat degradation (e.g., dam construction, pollution, or channelization), overexploitation, and introduction of invasive species [8,9,10,11]. For these reasons, wild specimens are difficult to obtain these days, and suitable fish for breeding are limited. Hatchery populations of *T. fulvidraco* are replenished with parents for artificial reproduction in each hatchery. In addition, there are few detailed records of the broodstock, making effective management impossible, and genetic variation in hatchery stock may be reduced [12]. The situation can become even more serious if artificial reproduction is carried out over multiple generations. Investigation of genetic diversity and management has so far been little considered for *T. fulvidraco*. Thus, the use of broodstocks collected from multiple points is recommended [12]. However, without detailed investigation, whether its genetic diversity is maintained remains a question.

Species intended for resource development must have high genetic diversity. However, the genetic diversity and genetic structure of wild populations of *T. fulvidraco* in South Korea have not been investigated. This study traced the origins of broodstock collected from wild populations at the time of collection from hatchery populations of *T. fulvidraco*. We also aimed to measure the genetic diversity of farmed and wild populations and provided baseline data for aquaculture strategies for future resource development. To test this hypothesis, we developed, tuned, and tested microsatellite DNA (msDNA) markers useful for population genetics studies.

## 2. Results

### 2.1. Genetic Diversity

Raw reads were obtained from each individual through whole-genome sequencing (Pf01: 6,231,011, Pf02: 5,705,048, Pf03: 5,123,581, and Pf04: 6,235,796; Appendix A). The assembly and screening results of the msDNA markers are shown in Appendix A. A total of 100 markers were screened, and the markers that were not amplified and did not show polymorphisms in each population were excluded (Appendix A). Multiplex PCR from the final 15 msDNA loci consisted of two sets (Appendix A).

The number of alleles (*N*_A_) ranged from 9 to 19, and the polymorphism information content (*P*_IC_) was >0.5, indicating appropriate marker selections (Appendix A). Genotyping using the MICRO-CHECKER software (ver. 2.2.3) showed no evidence of scoring errors or null alleles (Appendix A).

The 15 msDNA loci were analyzed for genetic diversity indices in the eleven populations (Table 1). The values described below are the average values of population genetic diversity information based on them.

*N*_A_ ranged from 7.60 to 11.00, the observed heterozygosity (*H*_O_) ranged from 0.771 to 0.873, and the expected heterozygosity (*H*_E_) from 0.735 to 0.844. We found that the KS and HT populations followed the Hardy–Weinberg equilibrium value (*P*_HWE_), but the remaining populations deviated from the *P*_HWE_. Inbreeding was observed only in SY, YD, and ND populations, but the inbreeding coefficient (*F*_IS_) was not significant. The ND population for resource establishment purposes showed the least genetic diversity with *H*_O_ = 0.771. The hatchery populations ASS and CT showed a high level of genetic diversity with *H*_O_ = 0.816 and *H*_O_ = 0.873, respectively.

### 2.2. Bottleneck Analysis

Using the infinite allele mutation (IAM) model, we identified significant bottlenecks in all populations (*p* < 0.05). Using the two-phase model (TPM), we identified bottlenecks in the ASS, KS, YD, and HT populations (Table 2). The KK population showed recent mode shifts, showing evidence of a bottleneck.

Among the eleven populations, the effective population size was 21–infinity. The effective population size of the ND population was infinity, and that of the CT population was the smallest at 21 (Table 2). The effective population size of the ND population for resource creation purposes was found to be infinity (Table 2). The AS, NW, PT, CT, and HT populations were smaller than the minimum effective population size of 100 required to prevent inbreeding depression.

### 2.3. Population Structure and Genetic Differentiation Analyses

The pairwise genetic differentiation (*F*_ST_) values of the msDNA datasets were all significant (Table 3), with the highest *F*_ST_ value between the AS and KS populations (*F*_ST_ = 0.156). The ASS, KK, KS, and YD populations exhibited a range of low *F*_ST_ values among populations (*F*_ST_ = 0.012–0.030). The AS, CT, NW, PT, HT, and SY populations also each exhibited a range of low *F*_ST_ values among populations (*F*_ST_ = 0.020–0.082).

Bayesian clustering analysis maximized the delta population constant (ΔK) value for population structure at *K* = 2 (Figure 1). At *K* = 2, the first group included the ASS, KK, KS, and YD populations. The second included the AS, CT, NW, PT, HT, ND, and SY populations. Although it did not inhabit the Nakdong River, the artificially produced ND population appeared to belong to the AS, CT, NW, PT, HT, ND, and SY populations.

Scatterplots of the discriminant analysis of principal components (DAPC) showed that the first populations were the ASS, KK, KS, and YD populations and the second populations were the AS, CT, NW, PT, HT, ND, and SY populations, which was the same result as that of STRUCTURE (Figure 2).

An analysis of molecular variance (AMOVA) for *T. fulvidraco* was performed on the eleven populations to determine the genetic structure (Table 4). The AMOVA, based on msDNA markers, showed 3.69% variation among populations within the groups, 7.10% variation among groups, and 89.21% variation within the populations.

## 3. Discussion

### 3.1. Genetic Diversity

Genetic diversity actively copes with changing environments and plays an important role in the persistence of species [13]. Genetic diversity at the species level in msDNA markers is generally indicated by H_O_ [14]. *Rita rita*, *Clarias gariepinus*, and *Pangasianodon hypophthalmus* are commercial fish species belonging to the same family of *Siluriformes* as *T. fulvidraco* [15,16,17]. The H_O_ range of *R*. *rita* was between 0.57 and 0.76, that of *C. gariepinus* was 0.43–0.61, and that of *P*. *hypophthalmus* was 0.42–0.50, suggesting that *T. fulvidraco* (H_O_ = 0.77–0.87) is a species with higher genetic diversity [15,16,17]. In fact, channel catfish *Ictalurus punctatus*, which is actively used commercially, had a H_O_ range of 0.72–0.77, which was similar to that of *T. fulvidraco* [18]. Therefore, *T. fulvidraco* has higher genetic diversity at the species level compared to other siluriform species, which may make it suitable as a commercial species.

In this study, hatchery populations (ASS and CT) showed H_O_ values ranging from 0.816 to 0.873, confirming their relatively higher genetic diversity than wild populations (YD, AS, ND, and SY). We observed that the genetic diversity of the wild populations ranged from 0.771 to 0.838, which was lower than that of the hatchery population (CT). The ND population is a single population released for the purpose of resource creation into a river (Nakdong River) where *T. fulvidraco* does not exist. The genetic diversity of the ND population was found to be the lowest at 0.771. In general, hatchery populations are reported to have low genetic diversity [15]. This is explained by the low number of broodstock used in hatcheries and the loss of genetic variation due to inbreeding [19]. However, in this study, the genetic diversity of two hatchery populations, ASS and CT, was relatively high. This may be because the Fisheries Resources Agency of South Korea recommends using diverse wild populations [20]. The low genetic diversity of the ND population intended for resource development may be of concern from a resource development perspective. However, it is believed that practical judgments can only be made if genetic diversity, as with the size of the effective population, is taken into consideration [13].

In the hatchery and wild populations, ND, SY, and YD had positive F_IS_ values, suggesting that inbreeding had occurred. However, since it was not significant, it was not judged to be an important implication. Excluding the ND, SY, and YD populations, eight populations (ASS, KK, KS, AS, NW, PT, CT, and HT) showed negative F_IS_ values, suggesting an influx of external populations, although this was not significant (*p* > 0.05).

Estimating the expected excess of heterozygosity in P_IAM_ is suitable for estimating recent bottlenecks [21]. The most recent population bottleneck in P_IAM_ was observed in all populations (*p* < 0.05). Habitat destruction reduces the population size and creates bottlenecks [13]. Wild populations have recently experienced population size declines, and active efforts are needed to conserve the resource.

N_e_ is important for species adaptation to changing environments and maintaining evolutionary potential [13]. Small population sizes can accelerate the risk of local extinction due to genetic drift and inbreeding effects [22]. Additionally, if populations are too small, managers should strive to preserve interconnected populations that are at least large enough to meet this minimum [23]. In this study, the four wild populations (AS, NW, PT, and HT) had a significantly low effective population size (N_e_ < 100). Frankham et al. [24] suggested that the N_e_ should be greater than 100 to avoid inbreeding depression in short-term generations. However, although the wild populations (AS, NW, PT, and HT) had N_e_ values below 100, there was an influx of external populations. For this reason, it is assumed that it was due to the release of fish from the captive breeding program of the local government. However, if this resource release is not maintained, they are likely to suffer from inbreeding depression in the short term. The effective population size of the ASS population, a hatchery population, was 127, which is higher than 100, but it is believed that the introduction of wild populations is necessary for continued resource expansion. The CT population from hatchery was found to have an effective population size of 21, which was relatively very low compared to the ASS population. It is considered important to improve the effective population size by introducing wild broodstocks from outside. The effective population size of the ND population for resource creation purposes appeared to be infinite, and it was confirmed to have a very large population. This means that the wild population was released only for the purpose of resource generation, and since it is presumed that the offspring of several broodstock populations were released, the effective population size is estimated to be very large. However, this may be due to the small number of samples; therefore, it was judged that a more detailed analysis would be necessary by increasing the sample size in the future.

### 3.2. Genetic Structure

In this study, we analyzed the genetic structure of eleven populations of *T. fulvidraco* that inhabit the Hantan River, Seomjin River, and Geum River, as well as the Nakdong River population formed for the purpose of resource creation (ND), and identified the origin of hatchery broodstock. The populations showed significant genetic differentiation in the msDNA dataset. Here, the genetic structure was divided into two groups (ASS, KK, KS, and YD vs. AS, CT, NW, PT, HT, ND, and SY). However, the results of this study are different from the pattern of genetic differences among river basins, mainly due to geographical differences. In general, freshwater fish in the Korean Peninsula show genetic differences depending on the geographical water system, and there is little genetic difference within the same river basin [25]. Interestingly, the PT, AS, and YD populations were geographically close to each other, so no differences in genetic structure were expected among them. However, it was confirmed that the YD population showed the same genetic structure as the ASS, KS, and KK populations. The first hypothesis is that the YD population may not have existed originally but became a stock-building population through release from a hatchery population. The second hypothesis is that this population was originally present in the area and that the released individuals may have been captured due to continued release from the hatchery. The first hypothesis is considered to have low probability because *T. fulvidraco* was continuously caught even before release [26]. The second hypothesis implies that since continuous fish offspring release was conducted in local government from 2017 to 2022, the fish seed population is likely to be captured. Although more samples are needed to investigate the YD population before release, it is necessary to secure more samples through future research and to perform sampling by season.

It was supported through STRUCTURE that the population for resource creation purposes was not genetically different from the AS, CT, NW, PT, HT, and SY populations. DAPC is a non-model method and shows higher reliability compared to STRUCTURE [27]. The DAPC results showed that the genetic structure of the eleven populations was divided into two groups (ASS, KK, KS, and YD vs. AS, CT, NW, PT, HT, ND, and SY). This shows a clear division, similar to the STRUCTURE results. Additionally, the *F*_ST_ values likewise supported significant genetic differences between the two groups. The hatchery populations ASS and CT were shown to have different genetic structures. Therefore, these significant differences in genetic structure can be important basic data for planning when creating resources [28].

The ND population is a group that did not originally exist in the water system but was created for the purpose of resource creation. Looking at the genetic structure results, it is suggested that the group was created from broodstocks of the AS, NW, CT, PT, HT, and SY populations.

According to the local government, the hatchery populations, ASS and CT, are believed to be produced from broodstocks of different river basins. This may explain the differences in genetic structure. It is presumed that the ASS population was mainly produced from broodstocks of the KK, KS, and YD populations, and the CT population is presumed to have been produced from broodstocks of the AS, NW, PT, HT, and SY populations. The use of broodstocks in hatchery populations is recognized as important. This requires consideration of the importance of genetic differences, genetic diversity, and effective population size upon release from the hatchery. Therefore, it is recommended that fisheries use an offspring release program (for local government), with an offspring population using KK, KS, SY, and YD populations with large effective population sizes as parents for conservation of the genetic diversity of the natural population in South Korea. In addition, we suggest using ND and SY populations for commercial aquaculture production.

## 4. Materials and Methods

### 4.1. Sampling and DNA Extraction

*Tachysurus fulvidraco* is used as an inland fisheries resource for commercial fishing and aquaculture form in the Republic of Korea; therefore, animal ethics approval was waived. As wild populations of *T. fulvidraco* were released from hatchery populations, we sampled the hatchery populations (ASS and CT). We also sampled wild populations using stocking records (KK, KS, YD, AS, NW, PT, HT, and SY) and sampled the ND population (for stock composition purposes). A total of eleven populations of *T. fulvidraco* were sampled (location and latitude–longitude details are provided in Figure 3 and Table 5). Each sampling was conducted between 2021 and 2023, and fin tissue from the fish was collected and soaked in 99% ethanol. Genomic DNA (gDNA) was extracted using the DNeasy Blood & Tissue Kit (QIAGEN, Hilden, Germany) according to the manufacturer’s instructions. The extracted gDNA was stored at −20 °C after dilution to 50 ng/μL to amplify msDNA loci.

### 4.2. Whole-Genome Sequencing and msDNA Marker Screening

Whole-genome sequencing was performed using a total of four individuals (Pfu01-Pfu04). It was performed using an Illumina MiSeq^®^ System (Illumina, San Diego, CA, USA) and included 300 bp paired-end library construction. For screening msDNA markers, contigs assembled from Geneious Prime ver. 2020.02 (Biomatters Ltd., Auckland, New Zealand) were selected for di-, tri-, tetra-, penta-, and hexa-sequences with more than five repetitions using the MISA tool [29]. Using Primer3 [30], we set the length of the appropriate primer to 17–25 bp, the size of the amplification product to 150–400 bp, and the melting temperature (Tm) value to 50–60 °C.

### 4.3. msDNA Genotyping

One hundred msDNA loci were randomly selected, and PCR was performed using an Applied Biosystems™ ProFlex™ PCR System (Thermo Fisher Scientific, Foster City, CA, USA). PCR was performed using 20 ng gDNA; 0.5 units AccuPower^®^ PCR PreMix (Bioneer, Daejeon, Republic of Korea); 0.5 μM forward primer labeled fluorescent dyes FAM, HEX, TAMRA, and ATTO565; and 0.5 μM reverse primer. PCR condition was as follows: pre-denaturation at 95 °C for 5 min, denaturation at 95 °C for 20 s, annealing at 55 °C for 20 s, and extension at 72 °C for 20 s. After 35 repetitions, the final extension was performed at 72 °C for 10 min, and the temperature was held at 8 °C. Each amplified PCR product was electrophoresed on 2% agarose gel to confirm the presence or absence and size of the amplified fragment. The PCR fragments were prepared by mixing a GeneScan™ 500 ROX size standard ladder (Thermo Fisher Scientific) and HiDi™ formamide and performing denaturation at 95 °C for 2 min, followed by termination at 4 °C. The allele sizes were determined using an Applied Biosystems™ ABI 3730*xl* DNA Analyzer (Thermo Fisher Scientific). Genotyping was performed using GeneMapper ver. 5 software [31].

### 4.4. Genetic Diversity Analyses

Scoring errors in msDNA loci were investigated using MICROCHECKER software (ver. 2.2.3) [32]. Genetic diversity was measured in terms of *N*_A_, *H*_E_, and *H*_O_ using CERVUS software (ver. 3.0) [33]. The *F*_IS_ and *P*_HWE_ analyses were performed using GENEPOP (ver. 4.2) [30] and ARLEQUIN software (ver. 3.5) [34]. Two methods were used to estimate bottlenecks. The methods involved the BOTTLENECK software (ver. 1.2.02) [35], a program for estimating bottlenecks through heterozygous excess testing, and IAM modeling [36]. A TPM and stepwise mutation model (SMM) [37] were used to estimate bottleneck, and TPM was performed with 10% variance and 90% SMM. In addition, each model had 10,000 iterations, and significance was verified using the Wilcoxon signed-rank test [38]. The *N*_e_ was determined using the linkage disequilibrium estimation of NeEstimator software (ver 2.1) [39].

### 4.5. Population Genetic Structure Analysis

ARLEQUIN software (ver. 3.05) [34] was used to analyze the differences in genetics between groups, as well as AMOVA. STRUCTURE software (ver. 2.3) [40] was used to perform genetic structure clustering analysis based on the Bayesian method model. To estimate the most suitable population, we set *K* to 1–10, and a suitable admixture model was applied to the mixture of water systems. The burn-in period was repeated 10 times with 10,000 iterations, and Markov chain Monte Carlo simulation was used with 100,000 iterations. To estimate a population-appropriate constant, we analyzed a study by the cluster results corresponding to *K* using STRUCTURE SELECTOR [41]. A discriminant analysis of principal components (DAPC) of the microsatellite dataset was performed on the population using the R package ADEGENET (ver. 2.1.3) [42], a non-model-based genetic clustering method.

## 5. Conclusions

*Tachysurus fulvidraco* is an important commercial fish species in the Republic of Korea. They are released from hatchery populations into wild populations through artificial propagation. The hatchery populations ASS and CT use broodstock collected from wild populations. In this study, 15 new msDNA markers of *T. fulvidraco* were found. We sought to identify the genetic diversity and structure of hatchery and wild populations and provide baseline data for useful resource development strategies. The observed heterozygosity in hatchery populations ranged from 0.816 to 0.873, and the observed heterozygosity in wild populations ranged from 0.771 to 0.840. The hatchery population CT had the highest genetic diversity but the lowest effective population size of 21. Supplementation of wild mothers is thus required to supplement the population size. The genetic rescue was divided into two groups with a STRUCTURE result *K* = 2. The first group included the ASS, KK, KS, and YD populations, and the second group included the CT, AS, NW, PT, HT, and SY populations. Therefore, it was confirmed that the ASS and CT populations originated from different populations. This study provides genetic information necessary for the future development and conservation of fishery resources for *T. fulvidraco*.

## Figures and Tables

**Figure 1 ijms-25-03923-f001:**
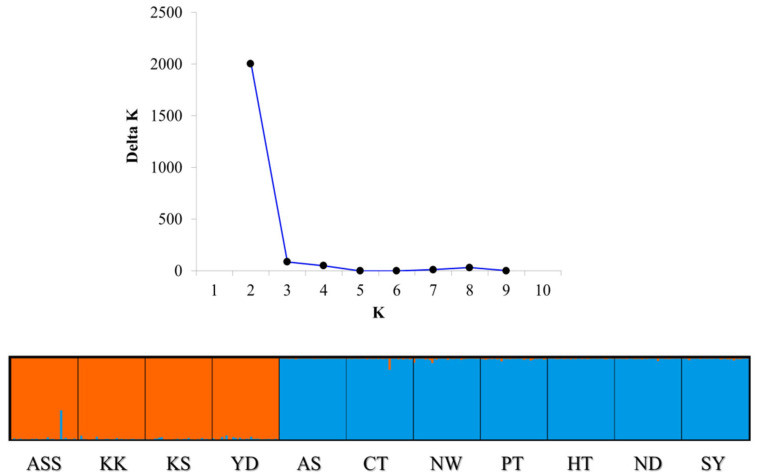
Population genetic structure of *Tachysurus fulvidraco* (*K* = 2). The ASS and CT populations are hatchery populations and the remaining populations are wild populations. The appropriate delta *K* information is represented for population constants. A single histogram represents the probability that an object and a particular color are assigned to a particular cluster.

**Figure 2 ijms-25-03923-f002:**
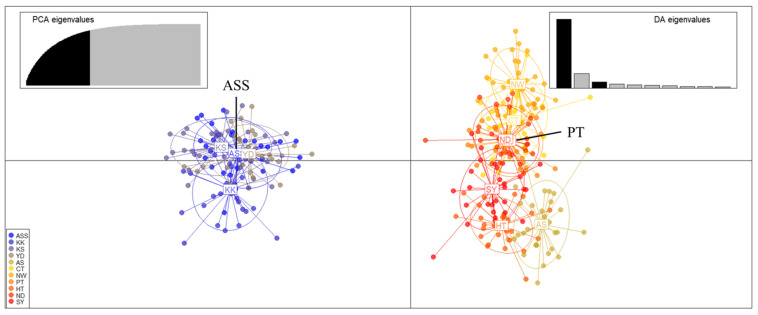
Scatterplots of discriminant analysis of principal components (DAPC) for *Tachysurus fulvidraco*. The colors shown on the plot are the population. Each color represents a different genetic cluster, and population abbreviations are provided for each cluster. The upper left graph represents the contribution of the eigenvalues of the selected principal components, and the upper right graph represents the variance explained by the eigenvalues of the two discriminant functions in the scatterplot.

**Figure 3 ijms-25-03923-f003:**
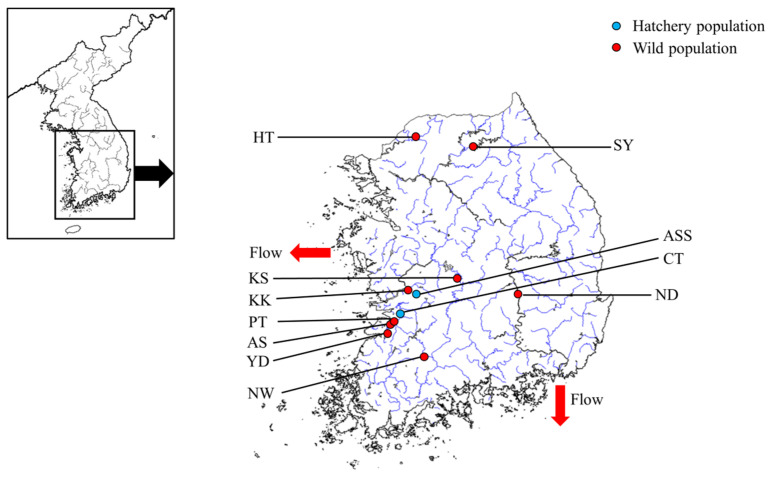
Sampling location of eleven populations of *Tachysurus fulvidraco*. Abbreviations for populations are given in Table 5.

**Table 1 ijms-25-03923-t001:** Average values of genetic diversity information for populations based on 15 microsatellite DNA loci of *Tachysurus fulvidraco*.

ID	Water System and Hatchery	*N*	*N_A_*	*H* _O_	*H* _E_	*P* _HWE_	*F* _IS_
ASS	Hatchery	30	9.93	0.816	0.813	0.000 ***	−0.003
KK	Geumg River	30	9.73	0.833	0.826	0.000 ***	−0.008
KS	Geum River	30	10.13	0.822	0.815	0.129	−0.009
YD	Yedang Reservoir	30	11.00	0.804	0.824	0.001 **	0.024
AS	Sapgyo Lake	30	7.60	0.791	0.735	0.000 ***	−0.078
NW	Seomjin River	30	9.60	0.838	0.802	0.000 ***	−0.045
PT	Pyeongtaek Lake	30	10.27	0.840	0.831	0.001 **	−0.012
CT	Hatchery	30	10.40	0.873	0.844	0.000 ***	−0.036
HT	Hantan River	30	8.27	0.820	0.791	0.051	−0.037
ND	Nakdong River	30	9.13	0.771	0.788	0.002 **	0.022
SY	Soyang Lake	30	10.27	0.802	0.822	0.029 *	0.024

*N*: number of samples; *N_A_*: number of alleles; *H*_O_: observed heterozygosity; *H*_E_: expected heterozygosity; *P*_HWE_: Hardy–Weinberg equilibrium value; *F*_IS_: inbreeding coefficient; * *p* < 0.05; ** *p* < 0.01; *** *p* < 0.001.

**Table 2 ijms-25-03923-t002:** Summary of information regarding bottleneck signature and effective population size for *Tachysurus fulvidraco* populations.

Population ID	*N*	Wilcoxon Sign-Rank Test				*N* _e_	(95% CI)
		*P* _IAM_	*P* _TPM_	*P* _SMM_	Mode-Shift		
ASS	30	0.000 ***	0.024 *	0.084	L-shaped	127	(74–377)
KK	30	0.000 ***	0.115	0.555	SHIFTED	212	(98–∞)
KS	30	0.000 ***	0.036 *	0.115	L-shaped	136	(73–628)
YD	30	0.000 ***	0.047 *	0.262	L-shaped	258	(109–∞)
AS	30	0.001 **	0.357	0.596	L-shaped	52	(35–91)
NW	30	0.000 ***	0.180	0.467	L-shaped	48	(37–69)
PT	30	0.018 *	0.756	0.972	L-shaped	79	(55–132)
CT	30	0.004 **	0.849	0.976	L-shaped	21	(18–25)
HT	30	0.001 **	0.005 **	0.047 *	L-shaped	32	(25–43)
ND	30	0.002 **	0.244	0.619	L-shaped	∞	(215–∞)
SY	30	0.000 ***	0.108	0.500	L-shaped	106	(68–226)

*N*: number of samples; *P*_IAM_: *p*-value of bottleneck test using infinite allele mutation model; *P*_TPM_: *p*-value of bottleneck test using two-phase mutation model (10% variance and 90% proportions of SSM); *P*_SMM_: *p*-value of bottleneck test using stepwise mutation model; *N*_e_: estimated effective population size by LDNe software; CI: confidence interval; * *p* < 0.05, ** *p* < 0.01, *** *p* < 0.001.

**Table 3 ijms-25-03923-t003:** *F*_ST_ among populations according to microsatellite DNA data of *Tachysurus fulvidraco*.

	ASS	KK	KS	YD	AS	CT	NW	PT	HT	ND	SY
ASS	-	0.000	0.000	0.000	0.000	0.000	0.000	0.000	0.000	0.000	0.000
KK	0.027	-	0.000	0.000	0.000	0.000	0.000	0.000	0.000	0.000	0.000
KS	0.030	0.027	-	0.000	0.000	0.000	0.000	0.000	0.000	0.000	0.000
YD	0.020	0.020	0.012	-	0.000	0.000	0.000	0.000	0.000	0.000	0.000
AS	0.142	0.146	0.156	0.145	-	0.000	0.000	0.000	0.000	0.000	0.000
CT	0.086	0.086	0.091	0.084	0.072	-	0.000	0.000	0.000	0.000	0.000
NW	0.109	0.108	0.111	0.099	0.077	0.020	-	0.000	0.000	0.000	0.000
PT	0.085	0.082	0.101	0.087	0.067	0.021	0.033	-	0.000	0.000	0.000
HT	0.104	0.099	0.106	0.098	0.082	0.033	0.057	0.038	-	0.000	0.000
ND	0.113	0.108	0.118	0.107	0.119	0.036	0.045	0.042	0.047	-	0.000
SY	0.091	0.090	0.103	0.090	0.073	0.032	0.047	0.019	0.020	0.028	-

Pairwise genetic differentiation significant level (above), *F*_st_: Pairwise genetic differentiation (below).

**Table 4 ijms-25-03923-t004:** Summary of information for analysis of molecular variance (AMOVA) for populations of *Tachysurus fulvidraco*.

Source of Variation	d.f.	Sum of Squares	Variance Components	Percentage of Variance	*F*-Statistics
Microsatellite DNA (Two groups based on the DAPC data (ASS, KS, KK, and YD vs. CT, ND, SY, AS, NW, PT, and HT populations)					
Among groups	1	168.472	0.48245	7.10	*F*_CT_ = 0.071 ***
Among populations within groups	9	189.940	0.25070	3.69	*F*_SC_ = 0.040 ***
Within populations	649	3934.400	6.06225	89.21	*F*_ST =_ 0.108 ***
Total	659	4292.812	6.79541	100.00	

d.f.: degrees of freedom; *** *p* < 0.001. *F*_SC_, *F*_ST =_ and *F*_ST_ were based on standard permutation across the full dataset.

**Table 5 ijms-25-03923-t005:** Details of the sampling information of *Tachysurus fulvidraco*.

Location	Code	Water System	Years	*N*	Location
Nonsan-si	ASS	Hatchery	2021	30	36°08′40″ N 127°04′35″ E
Nonsan-si	KK	Geumg River	2021	30	36°09′18″ N 127°00′19″ E
Geumsan-gun	KS	Geum River	2021	30	36°04′36″ N 127°34′30″ E
Yedang Lake	YD	Yedang Lake	2021	30	36°36′16″ N 126°47′50″ E
Sapgyo Lake	AS	Sapgyo Lake	2022	30	36°52′12″ N 126°50′30″ E
Namwon-si	NW	Seomjin River	2022	30	35°19′22″ N 127°18′41″ E
Pyeongtaek Lake	PT	Pyeongtaek Lake	2022	30	36°55′23″ N 126°58′31″ E
Jeollabuk-do	CT	Hatchery	2022	30	35°48′19″ N 126°50′54″ E
Yeoncheon-gun	HT	Hantan River	2023	30	38°00′53″ N 127°04′41″ E
Gumi-si	ND	Nakdong River	2023	30	36°06′48″ N 128°23′49″ E
Soyang Lake	SY	Soyang Lake	2023	30	37°55′27″ N 127°51′51″ E

*N*: number of samples.

## Data Availability

The original contributions presented in the study are included in the article/Appendix A, further inquiries can be directed to the corresponding author/s.

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
