# Peer review of "Genetic Structure and Diversity of Hatchery and Wild Populations of Yellow Catfish Tachysurus fulvidraco (Siluriformes: Bagridae) from Korea"

_ijms, 2024, doi:10.3390/ijms25073923_

Round 1

Reviewer 1 Report

Comments and Suggestions for Authors

In this study, 15 microsatellite markers for a freshwater fish species were developed. Overall, there are three points the authors should pay more attention to. Firstly, the manuscript presentation should be improved. Secondly, some statements should have literature supporting. Only 29 references are not enough for a scientific article, but maybe okay for a short communication. Thirdly, in the study of population genetic diversity analysis, genetic parameters will be stable only when the number of markers is more than 25. Supplement the number of SSR markers if possible. Here I list some examples where the authors should revise the manuscript. But remember these are only examples.

Line 3 Title: Move (Siluriformes: Bagridae) just behind Tachysurus fulvidraco.

The Abstract should be improved

Line 17 of which population the ”Fis” described here?

Line 27 When the scientific name first appeared, the name information should be added.

Line 29 When I checked the references 3 and 4, I found the data of 2022 had been published.

Lines 47-59: “Changes in catches of commercially...” References need to be added.

Line 69 T. fulvidraco is endemic to Korea? This is inconsistent with that in line 27 to 29.

Lines 149: “Inbreeding was observed only in YD and ND populations.” Inbreeding also occurred in SY population.

Line 193 delete the last comma

Lines 200-202: The meaning is not clear, please redefine it.

Line 211 Ho isn’t equal to genetic diversity.

Line 212 Just several wild populations, not all of them

Line 215 Italicize T. fulvidraco

Lines 300-301: Are the mothers of the ND population also the CT population? Indicate it clearly.

Lines 305-306: The use of parent in hatchery populations is thought to be associated with genetic differences, genetic diversity, and effective population size. Although the effective population size of ND population is large in the results, it has the lowest genetic diversity. In addition, the calculation result of effective population size of ND population may also be the reason for sampling. So why use ND as the parent for CT populations of hatchery?

A conclusion part is recommended.

Tables

I didn’t understand that was the genetic diversity information in Table 1 average means in the 15 loci? If so, the description for results should be more scientific.

Where is Table 2?

Author Response

Response author

Q1. In this study, 15 microsatellite markers for a freshwater fish species were developed. Overall, there are three points the authors should pay more attention to. Firstly, the manuscript presentation should be improved. Secondly, some statements should have literature supporting. Only 29 references are not enough for a scientific article, but maybe okay for a short communication. Thirdly, in the study of population genetic diversity analysis, genetic parameters will be stable only when the number of markers is more than 25. Supplement the number of SSR markers if possible. Here I list some examples where the authors should revise the manuscript. But remember these are only examples.

A1. Thanks for the review. We have improved the presentation and added references to some statements. Up to 39 references have been added.

At 9 or more microsatellite markers, the probability of assignment to parentage and offspring is as high as 99.99%. Therefore, 15 markers in this study were in LD and may demonstrate even genetic diversity across the genome. Additionally, we believe that these 15 markers were analyzed as a sufficient number of markers to be able to trace the ancestry of the offspring. Thank you for your understanding.

Zhang, J., Ma, W., Wang, W., Gui, J. F., & Mei, J. (2016). Parentage determination of yellow catfish (Pelteobagrus Fulvidraco) based on microsatellite DNA markers. Aquaculture international, 24, 567-576.

Q2. Line 3 Title: Move (Siluriformes: Bagridae) just behind Tachysurus fulvidraco.

A2. Thanks for the review. Edits were made to the manuscript.

Q3. The Abstract should be improved Line 17 of which population the ”Fis” described here?

A3. Thanks for the review. FIS values were deleted and corrected in the manuscript.

Q4. Line 27 When the scientific name first appeared, the name information should be added.

A4. Thanks for the review. Edits were made to the manuscript.

Q5. Line 29 When I checked the references 3 and 4, I found the data of 2022 had been published.

A5. Thanks for the review. Edits were made to the manuscript.

Q6. Lines 47-59: “Changes in catches of commercially...” References need to be added.

A6. Thanks for the review. Edits were made to the manuscript.

Q7. Line 69 T. fulvidraco is endemic to Korea? This is inconsistent with that in line 27 to 29.

A7. Thanks for the review. Edits were made to the manuscript.

Q8. Lines 149: “Inbreeding was observed only in YD and ND populations.” Inbreeding also occurred in SY population.

A8. Thanks for the review. Edits were made to the manuscript.

Q9. Line 193 delete the last comma

A10. Thanks for the review. Edits were made to the manuscript.

Q10. Lines 200-202: The meaning is not clear, please redefine it.

A10. Thanks for the review. Edits were made to the manuscript.

Q11. Line 211 Ho isn’t equal to genetic diversity.

A11. Thanks for the review. In general, HO stands for genetic diversity.

DeWoody, J. A., Harder, A. M., Mathur, S., Willoughby, J. R. The long‐standing significance of genetic diversity in conservation. Mol Ecol 2021, 30, 4147-4154. DOI: 10.1111/mec.16051

Q12. Line 212 Just several wild populations, not all of them

A12. Thanks for the review. Edits were made to the manuscript.

Q13. Line 215 Italicize T. fulvidraco

A13. Thanks for the review. Edits were made to the manuscript.

Q14. Lines 300-301: Are the mothers of the ND population also the CT population? Indicate it clearly.

A14. Thanks for the review. Edits were made to the manuscript.

Q15. Lines 305-306: The use of parent in hatchery populations is thought to be associated with genetic differences, genetic diversity, and effective population size. Although the effective population size of ND population is large in the results, it has the lowest genetic diversity. In addition, the calculation result of effective population size of ND population may also be the reason for sampling. So why use ND as the parent for CT populations of hatchery?

A15. Thanks for the review. Therefore, it is recommended that the fisheries seed release program (for local government), which is a seed population use KK, KS, SY, and YD populations with large effective population sizes as parents for conservation to genetic diversity of natural population in Korea. In addition, we suggest that the commercial aquaculture population use ND and SY populations.

Q16. A conclusion part is recommended.

A16. Thanks for the review. Edits were made to the manuscript.

Q17. Tables I didn’t understand that was the genetic diversity information in Table 1 average means in the 15 loci? If so, the description for results should be more scientific.

A17. Thanks for the review. Average values for the 15 loci analyzed in the population (HO, HE, FIS, number of allele). Edits were made to the manuscript.

Q18. Where is Table 2?

A18. Thanks for the review. Edits were made to the manuscript.

Reviewer 2 Report

Comments and Suggestions for Authors

Please correct:

Line No.

Abstract 11-20. I recommend improving the summary so that it is clear exactly what the goal is, who was used to achieve it, what the results and conclusions are. Do not use general words like, Hatchery populations, broodstock collected from the wild, Genetic structure DAPC results.

Introduction

24-41. This paragraph would be better written in two paragraphs or maybe three and include a macro that reviews the importance of aquaculture, because this is not directly related to work.

47-66. A lot of information is presented, without references. Some are correct and some are less or not clever enough. Any information in the introduction should be substantiated and if it is a hypothesis, it should be stated.

49-66. The purpose of the work should be clear and precise. If, for example, the goal is to compare wild populations with a domesticated population? Works that examined the subject should be reviewed in the introduction, and sections should be expanded. It should also be noted what the methods are to test for variation between populations.

Materials and Methods

67. There is a lack of basic information about the fish sampled both in the wild and those bred in aquaculture. Also the history and conditions of the habitat. I'm sure work has been done on the subject.

69-78. If it is possible to bring works describing the modeling that is done this way or to say that this is where the method was developed for the first time, that would improve.

82-106. There are two options. The first is that here an innovative method was developed and described for the first time, I guess that is not the case. Or there is a literature source that describes the method in detail and it should be cited. Results The writing of the results to be improved. Large tables should be transferred to the appendix at the end of the work.

Abbreviations must be defined precisely and try to make it easier to read, which is sometimes not found in the methods. (ND,CT, AS, NW, PT, CT, and HT).

The results should be written in such a clear way that the reader can understand the meaning of the findings without the need for tables (which are the basis), but the main findings will be clear. All analyzes (Figure 5) should be explained in the method and their meaning in the results in the context of this work.

Discussion

In my opinion, the discussion is not well written and mainly deals with clarifying the results or repeating them. Suggests that the authors rewrite it and keep the rules of writing a scientific paper. The results should not be repeated in the discussion, but should be compared to other work, and if not then this should be explicitly stated. Explanations that are hypotheses must be explicitly stated. And finally reach a conclusion which is the meaning of the work.

Author Response

Response author

Q1. Abstract 11-20. I recommend improving the summary so that it is clear exactly what the goal is, who was used to achieve it, what the results and conclusions are. Do not use general words like, Hatchery populations, broodstock collected from the wild, Genetic structure DAPC results.

A1. Thanks for the review. The wild and hatchery populations listed in the abstract are included to provide background information. Thank you for your understanding.

Q2. This paragraph would be better written in two paragraphs or maybe three and include a macro that reviews the importance of aquaculture, because this is not directly related to work.

A2. Thanks for the review. This paragraph was included to explain the background knowledge required for Tachysurus fulvidraco in aquaculture. Thank you for your understanding.

Q3. 47-66. A lot of information is presented, without references. Some are correct and some are less or not clever enough. Any information in the introduction should be substantiated and if it is a hypothesis, it should be stated.

A3. Thanks for the review. Edits were made to the manuscript.

Q4. 49-66. The purpose of the work should be clear and precise. If, for example, the goal is to compare wild populations with a domesticated population? Works that examined the subject should be reviewed in the introduction, and sections should be expanded. It should also be noted what the methods are to test for variation between populations.

A4. Thanks for the review. Edits were made to the manuscript.

Q5. 67. There is a lack of basic information about the fish sampled both in the wild and those bred in aquaculture. Also the history and conditions of the habitat. I'm sure work has been done on the subject.

A5. Thanks for the review. Edits were made to the manuscript.

Q6. 69-78. If it is possible to bring works describing the modeling that is done this way or to say that this is where the method was developed for the first time, that would improve.

A6. Thanks for the review. Edits were made to the manuscript.

Q7. 82-106. There are two options. The first is that here an innovative method was developed and described for the first time, I guess that is not the case. Or there is a literature source that describes the method in detail and it should be cited. Results The writing of the results to be improved. Large tables should be transferred to the appendix at the end of the work.

A7. Thanks for the review. Edits were made to the manuscript.

Q8. Abbreviations must be defined precisely and try to make it easier to read, which is sometimes not found in the methods. (ND,CT, AS, NW, PT, CT, and HT).

A8. Thanks for the review. Edits were made to the manuscript. Table 5 has been added. Because listing the group's abbreviations is too long. Thank you for your understanding.

Q9. The results should be written in such a clear way that the reader can understand the meaning of the findings without the need for tables (which are the basis), but the main findings will be clear. All analyzes (Figure 5) should be explained in the method and their meaning in the results in the context of this work.

A9. Thanks for the review. Edits were made to the manuscript.

[Figure 2. Scatterplots of discriminant analysis of principal components (DAPC) for Tachysurus fulvidraco. The numbers shown on the plot are the population. Each color represents a different genetic cluster and population abbreviations are provided for each cluster. The upper left graph represents the contribution of the eigenvalues of the selected principal components, and the upper right graph represents the variance explained by the eigenvalues of the two discriminant functions in the scatterplot.]

Q10. In my opinion, the discussion is not well written and mainly deals with clarifying the results or repeating them. Suggests that the authors rewrite it and keep the rules of writing a scientific paper. The results should not be repeated in the discussion, but should be compared to other work, and if not then this should be explicitly stated. Explanations that are hypotheses must be explicitly stated. And finally reach a conclusion which is the meaning of the work.

A10. Thanks for the review. Edits were made to the manuscript.

[Genetic diversity actively copes with changing environments and plays an important role in the persistence of species [13]. Genetic diversity of species-level in microsatellite markers generally refers to observed heterozygosity (HO) [14]. Rita rita, Clarias gariepinus, Pangasianodon hypophthalmus is a commercial fish species belonging to the same family of Siluriformes as T. fulvidraco [15-17]. The range of observed heterozygosity (HO) of Rita rita was between 0.57-0.76, Clarias gariepinus was 0.43-0.61, Pangasianodon hypophthalmus was 0.42-0.50, suggesting that T. fulvidraco (HO=0.77-0.87) is a species with higher genetic diversity [15-17]. In fact, channel catfish Ictalurus punctatus, which is actively used commercially, had an HO range of 0.72-0.77, which was similar to that of T. fulvidraco [18]. Therefore, T. fulvidraco has higher genetic diversity at the species level compared to other species, which may make it suitable as a commercial species.]

Round 2

Reviewer 2 Report

Comments and Suggestions for Authors In my opinion, after a careful check that all the changes that were made were inserted into the article as presented to me, the article should be accepted for publication in IJMS